# Sequestration within peptide coacervates improves the fluorescence intensity, kinetics, and limits of detection of dye-based DNA biosensors
Christopher M. Green[1,8], Deborah Sementa [2,8], Divita Mathur [3], Joseph S. Melinger[4], Priyasha Deshpande[5], Shana Elbaum-Garfinkle [5,6], Igor L. Medintz[1], Rein V. Ulijn[2,6,7] & Sebastián A. Díaz [1] ✉

Peptide-based liquid-liquid phase separated domains, or coacervates, are a biomaterial gaining new interest due to their exciting potential in fields ranging from biosensing to drug delivery. In this study, we demonstrate that coacervates provide a simple and biocompatible medium to improve nucleic acid biosensors through the sequestration of both the biosensor and target strands within the coacervate, thereby increasing their local concentration. Using the well-established polyarginine ($R_9$) – ATP coacervate system and an energy transfer-based DNA molecular beacon we observed three key improvements: i) a greater than 20-fold reduction of the limit of detection within coacervates when compared to control buffer solutions; ii) an increase in the kinetics, equilibrium was reached more than 4-times faster in coacervates; and iii) enhancement in the dye fluorescent quantum yields within the coacervates, resulting in greater signal-to-noise. The observed benefits translate into coacervates greatly improving bioassay functionality.

Detecting and reporting RNA and DNA is of great importance as they function as markers for viral, bacterial, and mycotic infections;[1] there is also growing thought that microRNAs can be utilized as biomarkers for a whole range of human diseases[2]. Complementary sequences serve as the most specific probes for these nucleic acids, able to detect single nucleotide mismatches and polymorphisms[3]. To this end, oligonucleotide probes, also known as molecular beacons (MBs), are very popular due to the ease of synthesis and range of modifications available[4]. Specifically, hairpin (HP) form DNA probes, composed of single-stranded DNA (ssDNA), which folds upon itself to form a stem-loop structure, are particularly useful in conjunction with energy transfer (ET) based fluorescent readouts[4,5]. Typically, MBs are labeled with two distinct dye species at the opposite ends for such a readout. In the closed state (HP form) of the MB, these dyes function

as highly efficient donor-acceptor ET system, primarily *via* Förster resonance energy transfer (FRET)[5]. Upon binding of the target, which is complementary to the loop and part of the HP stem, the MB unfolds and becomes a partially linear double stranded DNA (dsDNA); this separates the two fluorescent tags, avoids reformation of the closed HP form and minimizes the FRET, resulting in a large change in the fluorescent peak ratios of the dyes (Fig. 1A, B)[4,6]. While oligonucleotide-based MBs with much greater complexity have been reported[7,8], the straightforward HP form remains invaluable for testing nucleic acid biosensors in emerging applications, including coacervates.

Peptide-based in vitro liquid-liquid phase separated domains, i.e., liquid condensates, biomolecular condensates, or coacervates, are inspired by membraneless organelles with important regulatory functions in

[1]Center for Bio/Molecular Science and Engineering Code 6900, U.S. Naval Research Laboratory, Washington, DC 20375, USA. [2]Nanoscience Initiative at Advanced Science Research Center, Graduate Center of the City University of New York, New York, NY 10031, USA. [3]Department of Chemistry, Case Western Reserve University, Cleveland, OH 44106, USA. [4]Electronics Sciences and Technology Division Code 6816, U.S. Naval Research Laboratory, Washington, DC 20375, USA. [5]Structural Biology Initiative at Advanced Science Research Center, Graduate Center of the City University of New York, New York, NY 10031, USA. [6]Ph.D. Programs in Biochemistry and Chemistry, The Graduate Center of the City University of New York, New York, NY 10016, USA. [7]Department of Chemistry Hunter College, City University of New York, New York, NY 10065, USA. [8]These authors contributed equally: Christopher M. Green, Deborah Sementa.
✉e-mail: sebastian.diaz@nrl.navy.mil

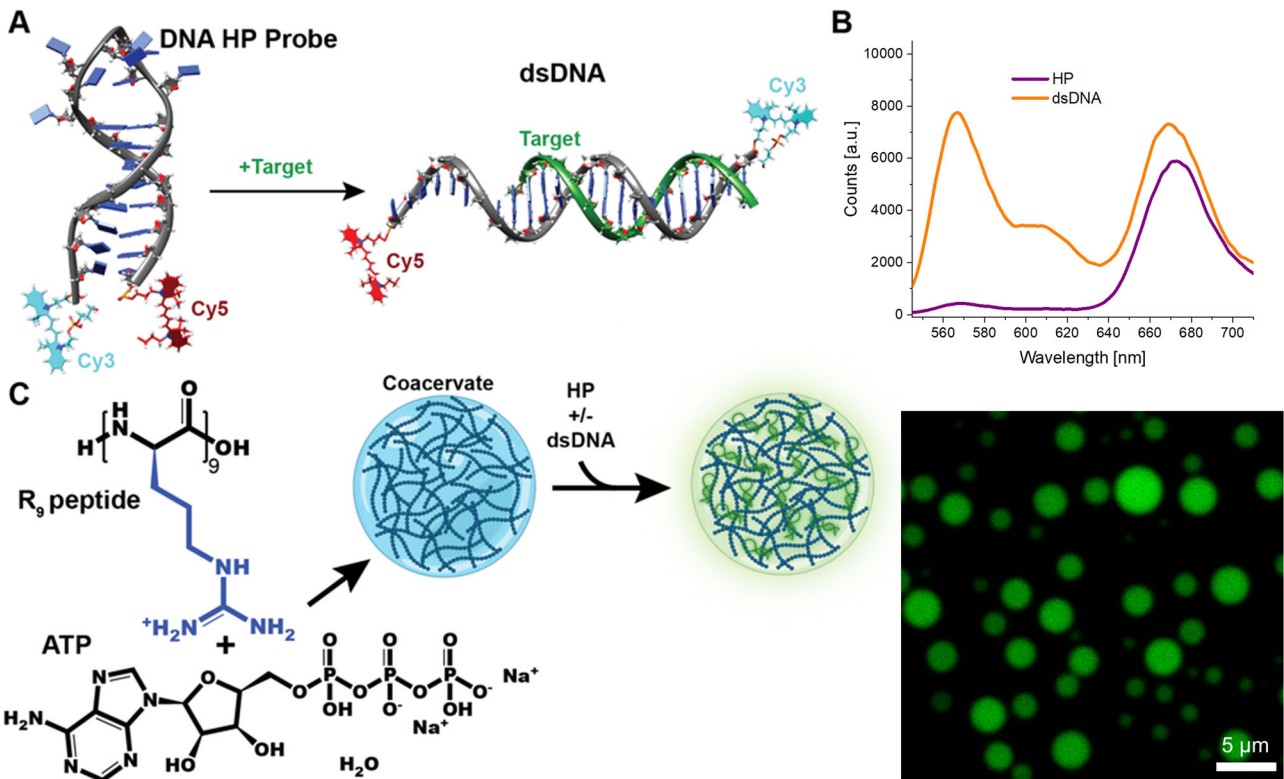

**Fig. 1 | Schematic of DNA and coacervates system used in this study (stuctures not to scale). A** DNA MB in the HP conformation, where addition of target transforms it into the dsDNA form. The MB is composed of a Cy3 donor and Cy5 acceptor FRET pair. **B** Fluorescence spectra of the two MB conformations.

**C** Combination of $R_9$ and ATP results in coacervate fomation. Addition of the MB allows for sequestration of the DNA into the coacervate, resulting in localization of the fluorescence within the coacervate as seen in the fluorescent micrograph at right.

biological systems[9]. Though the foundations of the field were set decades ago[10,11], coacervates as a de novo biomaterial has become a new focus due to the many exciting potential applications in fields ranging from biosensing to drug delivery[10,12–17]. Through these micrometer scaled self-forming membraneless structures, a liquid condensed phase rich in biomolecules separates itself from the original aqueous solution[18]. Coacervates are neither pure homogenous liquid phases nor heterogeneous aggregates, displaying varying degrees of supramolecular order inside a mostly disordered system[19,20]. The continuous aqueous environment inside and surrounding the coacervate enables steady exchange between these phases, a crucial factor when biosensing is the application of interest. Due to the condensed phase's need for charge equilibration, along with its modified hydrophobicity and dielectric constant, coacervates are capable of sequestering and concentrating molecules from solution[15,21]. In this context, simple peptide-derived coacervates have been reported to successfully recruit nucleotides into the condensate phase[21,22], similarly to proteins that often enlist DNA or RNA within membraneless compartments inside the cells[23,24].

Here, we found that coacervates are a minimalist and biocompatible way to increase the sensitivity of nucleotide biosensors as they can concentrate both the MB and target strands within the coacervate. We combined the polyarginine peptide, specifically $R_9$, as a positively charged component with adenosine triphosphate (ATP), as the negatively charged counterpart that co-assembles forming the liquid droplets[25,26]. At concentrations greater than 250 µM of the peptide and 350 µM of ATP[27], coacervates form with diameters ranging between 1–4 µm. These coacervates efficiently capture the majority of the DNA-based MB, as verified by fluorescence microscopy (Fig. 1C).

The change in the fluorescence peak ratios of the MB is directly correlated to the distribution between the HP and dsDNA forms, which is a function of the concentration of the target strand in solution[6]. The oligonucleotide sequestration within the coacervates resulted in higher localized concentrations and therefore increased sensitivity. Our experiments have

found a greater than 20-fold lowering of the limit of detection (LOD) within coacervates when compared to reactions in bulk buffer solutions. Furthermore, we reported additional benefits of utilizing coacervates including: (i) an increase in the sensing kinetics with the final equilibrium stage being reached more than 4-times faster in coacervates; and (ii) an increase in the dye fluorescent quantum yields (QYs) within the coacervates, resulting in greater signal-to-noise (S/N) ratios.

## Results and discussion
### Coacervate design
To test our proof-of-concept system, we used the well-characterized poly-arginine ($R_9$) – ATP coacervate self-assembled in a low ionic strength buffer of 10 mM Tris·HCl, 15 mM KCl, 0.5 mM $MgCl_2$, pH 7.6 buffer (when we utilize the term 'Buffer' condition it refers to this same buffer.)[25,26,28]. Unless otherwise noted, the $R_9$ was added to the buffer (final concentration 400 µM), followed by ATP (final concentration 750 µM); the consequent macroscopically observable cloudiness confirmed the formation of the coacervates[12]. We note that perfect charge balance is not achieved through the macromolecules with this mixture; this was done purposefully. We chose to keep the charge relatively positive (~4+:3−) to maximize DNA sequestration, though counterions will balance the overall net charge of the phases[29]. The DNA MB was then added (250 nM final concentration) to the liquid system and left to equilibrate for 20 min at RT (20 °C) before measurements were undertaken.

### MB design and characterization
As an initial investigation, the nucleotide biosensor has been based on a simple design of a double-labeled DNA HP MB that senses a target strand (Table 1). The melting temperature of the HP form of the MB was determined to be 53 ± 1 °C in the buffer (coacervate values were not determined as the coacervate is not stable at these higher temperatures), while the dsDNA was determined to be 48 ± 2 °C (See supplementary data Fig. S1). As

## Table 1 | DNA sequences for MB and target

| Strand | Sequence (5'→3') |
|---|---|
| MB | /5Cy5/CTACTATT**TTGATGAGATAGTAGA**/3Cy3/ |
| target | TCTACTATCTCATCAA |

The bold sequence highlights the complementary section of the MB sequence to the target. Control samples with the MB design, where only one of the two dyes are present, were also purchased and were used for the Cy3 and Cy5 only determinations.

## Table 2 | Fluorescence quantum yield of the dyes on the MB in varying confirmations (HP and dsDNA) and conditions (buffer and coacervate)

| | Buffer | | Coacervate | |
|---|---|---|---|---|
| | HP | dsDNA | HP | dsDNA |
| Cy3 donor | 0.199 ± 0.01 | 0.16 ± 0.01 | 0.52 ± 0.04 | 0.62 ± 0.04 |
| Cy5 acceptor | 0.20 ± 0.01 | 0.28 ± 0.01 | 0.39 ± 0.03 | 0.36 ± 0.02 |

Determinations based on secondary standards (rhodamine B and oxazine 720)[30], uncertainties are based on propagating uncertainties from standards and experimental spectroscopy.

we worked consistently at 20 °C, we ensured that the initial structure was stable and that upon binding of the target it would not subsequently be released.

It is important to have an understanding of the predicted fluorescent changes in our MB system, and to that end, we considered the parameters of the MB DNA as well as the FRET pair. The hybridized MB:target pair would have 16 base pair (bp) in dsDNA form (5.4 nm); the full length of the MB is 24 bp plus the dye linker lengths, so we could expect separations between the dyes, $r_{DA}$, of up to ~9 nm if it was fully linear. The FRET efficiency ($E_{FRET}$) would depend on the dye properties, particularly the change in Förster distance, $R_0$, as a function of shifts in spectral overlaps and fluorescent QYs. In Table 2 we provided QY values for the dyes in the HP or in the linear (dsDNA) format, as well as in the coacervate, and compared them to the buffer alone.

When the MB is in the HP form, the dyes are in close proximity, leading to high $E_{FRET}$; the dyes are, in fact, vibronically coupled and interacting strongly, as suggested by the shifts in the absorbance spectra (supplementary data Fig. S2), and thus we can assume ET saturation[30,31]. We note that this strong coupling has been repeatedly shown to lower the fluorescence QY of cyanine dyes, including of heterodimer pairs as in this case[31,32]. The proximity of specific nucleotides can also have an effect, as seen in the different dye QYs in the HP and dsDNA forms in Table 2. It is interesting to note that there is a shift and redistribution of the absorption bands in HP form when comparing the buffer to coacervate (supplementary data Fig. S2). This suggests the dyes are in a different dielectric and polar environment as well

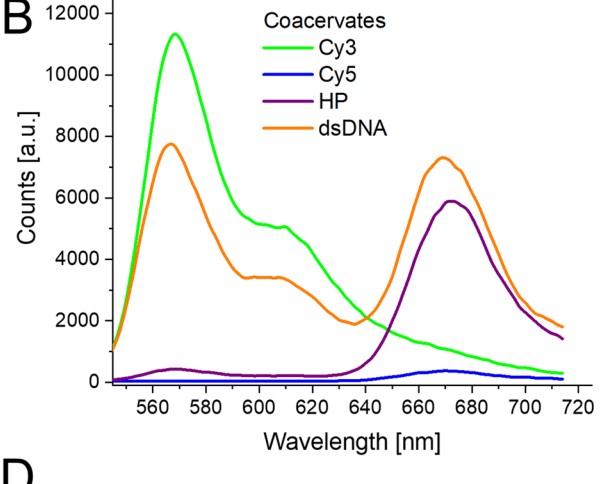

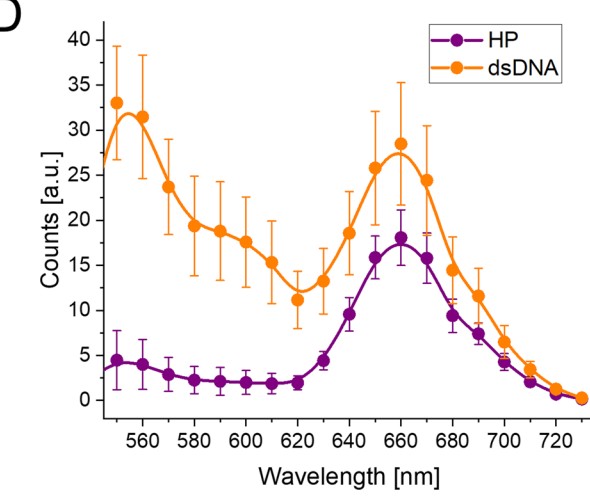

**Fig. 2 | Fluorescence under different conditions. A** Fluorescence emission of DNA MB in aqueous buffer, variations of design (Cy3, Cy5, and both dyes) and conformation (HP and dsDNA). Excitation 520 nm. **B** Fluorescence emission of DNA MB in combination with R₉-ATP coacervates, variations of design and confirmation. Excitation 520 nm. **C** Confocal fluorescence micrograph of R₉-ATP coacervates with dsDNA form MB sequestered into the coacervates. Excitation 520 nm, detection 540–740 nm. **D** Fluorescence spectra obtained from confocal fluorescence microscopy of coacervates with the two MB conformations. Uncertainties arise from averaging the spectra of over 10 different images and region of interest (ROI) for each conformation.

as likely change their relative position/orientation substantially in the coacervate, which could be due to the complex phase diagrams of the coacervates[33]. Due to their strong coupling in the HP form, FRET assumptions are likely invalid (a short primer on FRET theory is available in Supplementary Note 1, with greater detail available in the references)[32,34], therefore the $R_0$ values are more relevant for the FRET pair in the dsDNA form. For buffer the $R_0$ value is 5.5 ± 0.3 nm, while within coacervates it is 6.8 ± 0.3 nm, the increase being principally due to the increased QY of the Cy3. Estimating the $r_{DA}$ based on the physical properties of the DNA, a range of 6–9 nm would be reasonable. This broad range arises from the fact that even upon hybridization with the target, part of the MB remains as ssDNA, and that the dye linkers are very flexible. As shown in Fig. 2, considering the fluorescence spectra of the MB in both states (HP and dsDNA) as well as the individual dye components, the Cy3 quenching went from 95% to 45% in buffer conditions and from 96% to 32% within coacervates[32]. Using the steady-state spectra to determine the $E_{FRET}$ based on the Cy3 quenching, we confirmed that the experimental $r_{DA}$ in the dsDNA conformation was 5.6 ± 0.5 nm in the buffer and 7.7 ± 0.6 nm in the coacervate, which are both values within the predicted distances. This is important as it confirms that the MB is undergoing the expected conformational changes both in the buffer solution and in the more viscous and complex coacervate interior. Unexpectedly, we observed that the Cy5 signal increased upon going from the HP to dsDNA form (See Fig. 2). This is due to the above-mentioned self-quenching of Cy5 by the vibronic coupling in the HP form[30–32]. Therefore, though the $E_{FRET}$ is greatly reduced going from HP to dsDNA form, the fluorescence QY of both dyes improves to compensate for this issue. Nevertheless, the current MB design still functions for our proof-of-concept study while future MBs can alternatively be designed to avoid this issue.

## MB properties upon coacervate sequestration

Fluorescence microscopy confirmed the recruitment of the DNA biosensor inside the liquid peptide coacervates (Fig. 2C). For these experiments, we added either the MB by itself or the dsDNA form of the sensor (target strand was added in a 1:2 ratio of MB:target, the excess target assures that all the MB was in the dsDNA form), previously combined and annealed. It was observed that the fluorescence was almost fully localized within the coacervate droplets (>99%, estimated by comparing coacervate fluorescence intensity to background intensity, supplementary data Fig. S3), spectral detection showed the expected Cy5/Cy3 ratio (Fig. 2D, and supplementary data Fig. S4). Then, we exploited the fact that coacervates can be concentrated through centrifugation[35]. We added MB to the coacervates and then subsequently centrifuged the sample (10 min at 27k RCF), supernatant fluorescence compared to an uncentrifuged sample was less than 0.3 ± 0.1%,

while an equivalent experiment in just buffer showed no decrease in fluorescence (supplementary data Fig. S5). Through these two experiments, we confidently confirmed that more than 99% of the DNA was sequestered into the coacervates. Additionally, fluorescence recovery after photobleaching (FRAP) experiments corroborated the droplets' dynamic nature, independently of the DNA being in the HP or dsDNA form (supplementary data Fig. S6).

We observed a higher fluorescence QY of the dyes within the coacervates, resulting in a stronger signal from the MB (Table 2). In addition, fluorescence lifetime (τ) analysis validated the steady-state results (Fig. 3A–C). Specifically, the Cy3 has the greatest increase in QY and lifetime (~2.6-fold) while the Cy5 has only a slight increase (~25%). This enhancement seems to arise from the greater viscosity found within the coacervates[36], which possibly limits the rotation of the methine bridge of the cyanine dyes, a common non-radiative decay mechanism[37,38].

## Biosensing within coacervates

We then proceeded to test the MB capability to function as a biosensor within the coacervates. In contrast to the data presented so far, where the HP and target DNA were combined before adding them to the coacervate or buffer solution, in subsequent experiments we added the MB to the coacervate and then afterwards added the target strand to the reaction volume, exploiting in this way the dynamic interface of the coacervates. We noted that control experiments with a random DNA sequence added instead of the target resulted in no change in the MB fluorescence (supplementary data Fig. S7).

With the aim of testing the effect of individual coacervate components on the ability to interact with the MB sensors, we realized an experiment where only one of the two elements, either the $R_9$ or the ATP, was added to the buffer solution and then the MB was added. While the ATP only had no effect, the $R_9$ appeared to interact with the MB and precipitate or quench the dye-labeled DNA, due to charge interactions. Thus, keeping constant the ATP concentration at 800 μM, we modified the amounts of $R_9$ added, with the lowest amounts being below the critical concentration levels needed to form the coacervate (Fig. 4A). Precisely, under 250 μM the peptide quenches/precipitates the MB (mechanism unclear at this time, though electrostatics is likely driving the interaction). Starting at that concentration, we see coacervate formation and a recovery and then improvement of the fluorescence signal. We then added either 10 μl of water (no target) or 300 nM of target DNA, and followed the spectra on a Tecan fluorescent plate reader. Figure 4B shows the time traces of the change in Cy5/Cy3 ratio with the different $R_9$ concentrations. As we are following the normalized acceptor/donor fluorescence ratio, the smaller the value the greater the signal change, i.e., a value of 1.0 is no change while the closer the value is to

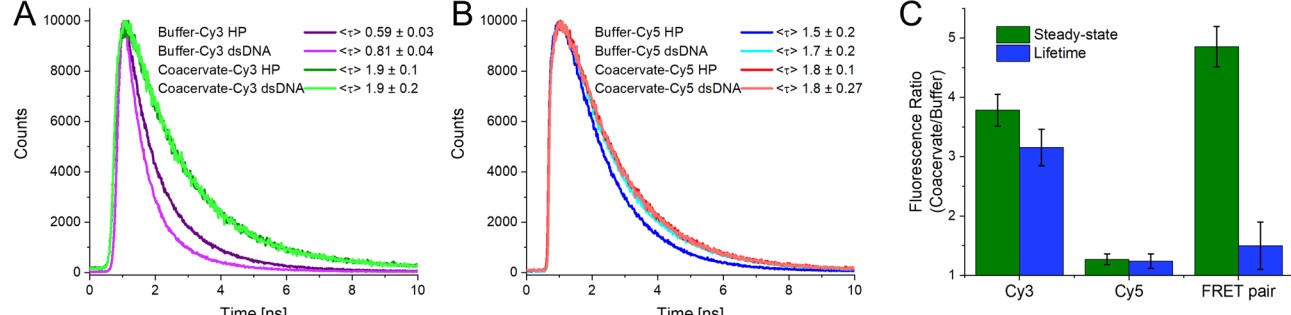

**Fig. 3 | Comparative fluorescence QY of dyes in buffer and coacervate.**
**A** Fluorescence τ of Cy3 dye in the HP and dsDNA confirmation and in buffer or with coacervates. τ determined through multiexponential tail-fittings, presented as amplitude averages. **B** Fluorescence τ of Cy5 dye in the HP and dsDNA confirmation and in buffer or with coacervates. **C** Increase in steady-state fluorescence emission and fluorescence lifetime of Cy3, Cy5, and DNA containing both dyes (FRET pair) when comparing the coacervate condition to the buffer condition. The numerical

result arises from the average of both the HP and dsDNA confomation. Uncertainties represent propogation of the uncertainties of lifetime fits and the intensities of the individual Cy3 and Cy5 components. The FRET pair, which looked at the total fluorescence from both dyes, has a noticeable difference between steady-state and lifetime increases, we assign this to the fact that the fluorescence lifetimes will under-represent quenched dark states of the dyes, particularly prevalent in the HP form.

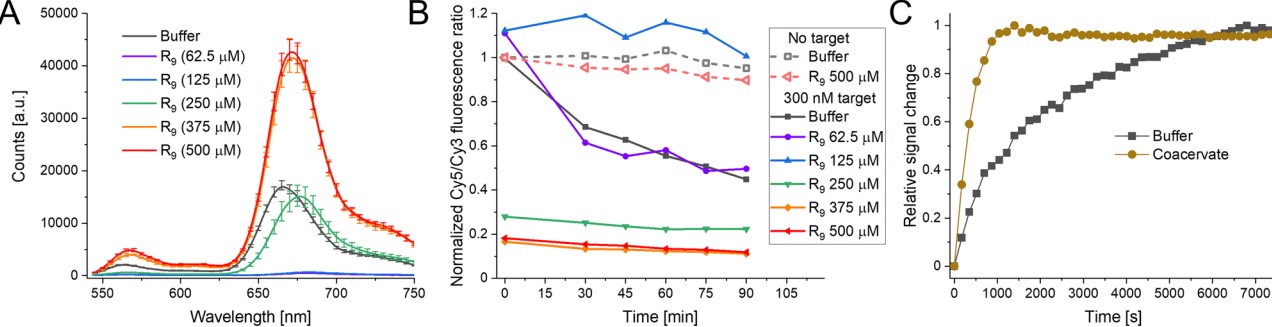

**Fig. 4 | Fluorescence of MB in the presence of varying concentrations of $R_9$ upon addition of target strand. A** Fluorescence of MB in HP confirmation with varying amounts of $R_9$ peptide in solution that cover above (≥250 μM-green line) and below (≤125 μM-blue line) the concentration required for coacervate formation. This can be observed by the large drop in detected fluorescence in the lower concentrtions.

Uncertainties are the results of the sample deviation from three repeats. **B** Time trace of Cy5/Cy3 fluorescence ratio upon addition of 300 nM target strand in varying concentrations of $R_9$ peptide (ATP constant at 800 μM). **C** Time trace of the relative Cy5/Cy3 ratio (i.e., signal) change of a 1:1 MB:target reaction in buffer or coacervate (normal concentrations) conditions.

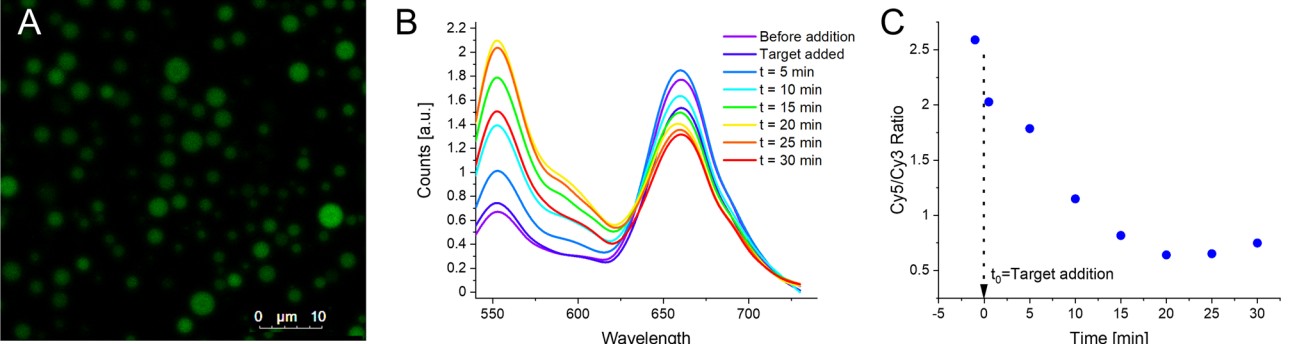

**Fig. 5 | Following biosensing on a microscope. A** Confocal fluorescence microscopy image of $R_9$-ATP coacervates with HP form MB sequestered into the coacervates. Excitation 520 nm, detection 540–740 nm. **B** Fluorescence spectra obtained from confocal fluorescence microscopy of coacervates seen in A at different times

upon addition of a 1:2 ratio of MB:target. Spectra are obtained by averaging over the entire field of view. **C** Time trace of Cy5/Cy3 fluorescence ratio of confocal image seen in **A**.

0.0 the more target was detected. We found that the change was greater in the coacervate samples, with 0.2 final ratio in the coacervates as compared to 0.5 in the buffer. The low (below coacervation concentration) $R_9$ concentration samples either completely failed ($R_9 = 125$ μM) or worked equivalently as the buffer, but with much lower fluorescence, i.e., lower S/N ratio in the $R_9$ 62.5 μM sample. We hypothesize that, in the $R_9$ 62.5 μM sample, there remains a small amount of DNA that did not interact with the peptide and thus was able to function as a MB.

One interesting observation from Fig. 4B was that, not only was the signal stronger, but it appeared to reach the final state much faster within coacervates. In Fig. 4C we show a representative time trace of the signal change of the MB recognizing the target strand, these experiments were realized following only the peak intensities allowing for much denser surveying of the initial time points. The y-axis has been normalized to the total fluorescence ratio change of each sample and even still, the coacervate sample has a ~4-times shorter response time. We noted that though kinetics were consistently faster for the coacervate, the relative difference was variable. In order to compare kinetics, seven different experiments were conducted and we observed an increase in speed ranged from 2 to 6.5 times faster in the coacervates, with a calculated average increment of 4.4 ± 1.7 times faster. Regarding this, slight variations in concentration, ambient temperature, mixing, and other variables that could modify solution viscosity or coacervate diameters, for example, may be responsible for this variation[39].

To confirm that the target DNA was being taken in by the coacervates containing the MB, we replicated experiments from the plate reader on the microscope. Coacervates were added to a functionalized slide (See Methods Section) in order to reduce their coalescence on a glass surface, and imaged

using 520 nm excitation (Fig. 5A)[16]. The spectral information obtained from the liquid droplets showed the expected small Cy3 emission and large Cy5 emission. We then added the target strand in twofold excess to the MB and continued to image the coacervates. As expected, the fluorescence ratio began to shift towards the Cy3 due to the switch from the HP to the dsDNA form (Fig. 5B, C). We were able to observe that at approximately 20 min after addition the reaction reached its maximal signal.

We initially did not expect the improvement in S/N due to the enhanced fluorescence, nor the improved kinetics that the coacervates would supply. Though these were welcomed observations, we wished to test our initial hypothesis. We hypothesized that, due to the sequestration of the DNA and the increased local concentration we should be able to improve the sensitivity of nucleotide biosensors. Simple mass estimates based on solution concentrations predicted that the coacervates (the $R_9$, ATP, and DNA) composed <0.1% of the solution mass. Even if we assume this increases an order of magnitude when considering solution volume, we have still concentrated our DNA 100-fold, which would be on the low end of the 40000-fold reported by Liu et al. (we note that the coacervates formed by Liu et al. were based on a more complex polymer-oligopeptide hybrid)[21]. To test this we realized triplicate measures of the Cy5/Cy3 ratio as a function of the amount of target strand added to the MB. As can be observed in Fig. 6, the MB within coacervates worked much better, i.e., smaller Cy5/Cy3 ratio, than the buffer only system (representative raw data shown in supplementary data Fig. S8). In fact, using the classical LOD estimate based on 3σ of a blank (0 target added) which resulted in the LOD being a ratio change below 0.97, the coacervate could detect down to 0.005 target per HP (Fig. 6B), while the buffer only system had a LOD between 0.1 and 0.25 target per HP. This is a greater than 20-fold increase in sensitivity.

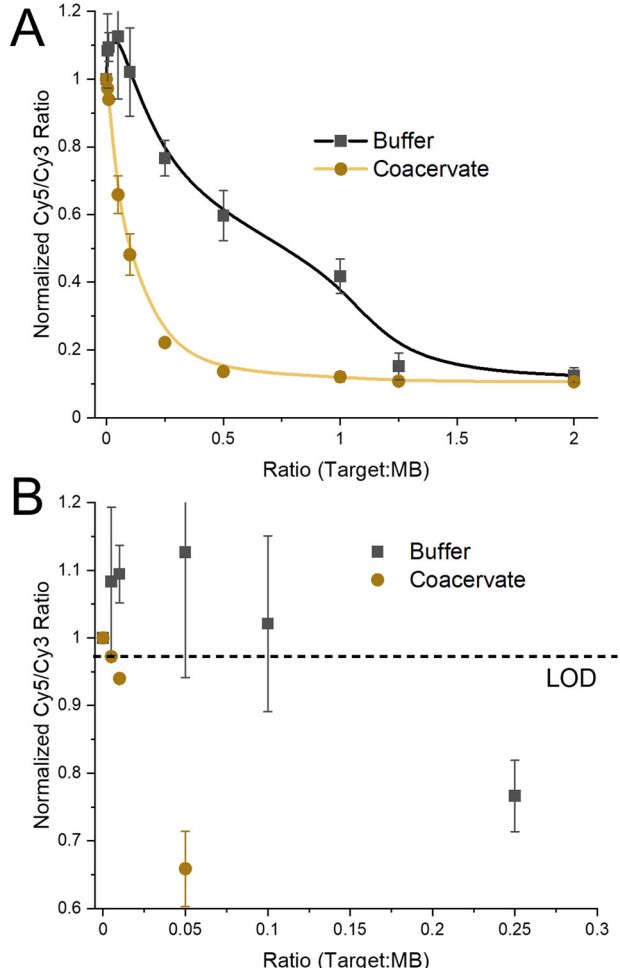

**Fig. 6 | Comparison of MB functionality in buffer and coacervate. A** Cy5/Cy3 fluorescence ratio as a function of the amount of target strand added to the MB in HP form (represented as a ratio of Target:MB). Uncertainties are the results of the sample deviation from three repeats. **B** Zoomed in data set from **A**, with the LOD marked as a dashed line.

The coacervate system is much more sensitive, and importantly due to the increased S/N, is able to distinguish between small amounts of target added, down to 0.5% of the MB concentration. The buffer system is much noisier (estimates from the data in Fig. 6 provide an ~8-fold increase in uncertainty), and in fact due to the changes in QYs, the initial addition of target appears to actually increase the Cy5/Cy3 ratio, only somewhere between the 0.1 and 0.25 target:MB ratio does the change in fluorescence fall below the LOD. The coacervate system does saturate the signal change sooner, signifying that the buffer system might be preferable for distinguishing between higher amounts of target strand.

## Conclusions

We have shown the use of coacervates as a medium for increased functionality of oligonucleotide-based biosensors. Though there has been evidence in the literature that coacervates are capable of sequestering nucleotides[21,23,24], our work demonstrates their capability to use toehold-mediated displacement for biosensing within the coacervates in line with what is seen in common buffers. Yet, the functionality of DNA biosensors was not only preserved within coacervates, but they also worked at much lower concentrations and proportions of target strand to MB strand, due to the droplets capability to sequester oligonucleotides, raising their local concentration more than 100-fold. It is also possible that the R9 destabilizes the MB allowing for a lower energy barrier for transition to the dsDNA form[40,41], improving kinetics and sensitivity. Other benefits included the

improved S/N due to the increased fluorescence intensity of the dyes and enhanced kinetics. Though the full cause of the increased kinetics is not known at this time; the local increase in concentration maximizing the collision frequency, and R9 destabilization of the MB, may play the two largest roles.

While we feel confident that the DNA is sequestered within the coacervates and our system appears homogenous based on our experimental evidence (microscopy and FRAP), anisotropic domains within the coacervate, liquid-crystal domains, or heterogonous probe distribution may be possible[11,33,42]. There are examples in other works of dye localization to the phase interface between the coacervate and the continuous buffer[21], and modified distributions could affect the biosensing response due to changes in local concentration or accessibility.

We do recognize that the system may have limitations. While coacervates could be tailored to different conditions, employing for example other polypeptide-based coacervates for DNA sequestration, there are limits on the amount of charges that can be added to maintain the stability of the system. It is therefore possible that at higher end concentrations or in very dense sample matrices, the coacervates could fall apart, negating the benefits[13]. Furthermore there are limits to temperature stability as well, for our system we noticed coacervate degradation (followed as a decrease in the scattering signal) at temperatures above 35 °C, with complete disappearance at 45 °C. We note that as part of a follow-up study we have tested the addition of enzymes to the mixture, from ~1–100 nM, and found no decrease in coacervate stability and efficiency. It is also possible that not all dyes will have the added benefit of increased fluorescent QY within the coacervates. Recent reports of autofluorescent coacervates, these were different peptide sequences and autofluorescence has not been observed in R9, could also interfere in the detection of the signal[43]. The extensive variety of peptide structures known to form coacervates and the varied nucleotide MBs (e.g. different designs, alternative nucleotide chemistries, wide-ranging dyes) should allow solutions to be found for most of these issues[12,20,27].

While we already report considerable improvements upon comparing DNA biosensor performance in coacervates vs buffer, we believe even greater enhancement could still be achieved. For example, the designed MB could be improved by avoiding the quenching of the Cy5 in the HP form and by extending the dsDNA form to lower the FRET in the open form and maximize the Cy3 signal. This would increase the dynamic range of the sensor by creating a much greater change in the ratio of Cy5/Cy3 fluorescence and would likely improve the LOD even further. It is also possible that further optimization of the polypeptide chosen for the coacervate could lead to greater sequestration (lower LOD), smaller coacervates (faster kinetics), and more viscous interiors (improved S/N). As seen in the initial experiments, the MB can already have interacted with the target and subsequently integrate into the coacervate to improve the signal, this means that the order of addition is not important, which should simplify sample preparation for biosensing applications. Beyond biosensing, DNA systems are being postulated as possible next-generation data storage and computing hardware, but have been challenged by low retrieval and reading speed[44,45]. To date most of the approaches to realize these applications utilize similar recognition and displacement approaches as shown within this manuscript, it thus seems conceivable that coacervates will provide a way to improve the speed and minimize the leakiness of these systems.

## Materials and methods

R9 peptide (used after trifluoroacetic acid, TFA, removal) and DNA were purchased from Genscript and IDT, respectively. ATP (CAS: 34369-07-8) and salts for buffers purchased from Sigma Aldrich and used as is. The utilized buffer was 10 mM Tris·HCl, 15 mM KCl, 0.5 mM MgCl2, pH 7.6.

### TFA removal

Peptides were dissolved in a 50 mM HCl solution to a concentration of 1 mg/ml. The solutions were sonicated for 5 min and lyophilized. The procedure was repeated 5 times.

## Slide surface functionalization

Glass coverslips were surface coated following the method previously described[16,26]. Briefly, cover slides were treated with 0.5 M KOH in iso-propanol for 30 min, washed and dried overnight at 90 °C. Subsequent silanization was carried out using a 3 mg/ml solution of N- (Triethoxysilylpropyl)-O-polyethylene oxide urethane in toluene for 4 h. Slides were dried and wiped to remove any evaporation-induced marks. Functionalized slides were used for fluorescence imaging as it allows the condensates to adopt more spherical shapes[26].

## Sample preparation

Coacervates were formed by creating stock solutions of $R_9$, ATP, and the required DNA, along with the buffer solution. $R_9$ was added to the buffer in an Eppendorf tube and then ATP was added subsequently and mixed through pipetting. In general the MB, or the pre-formed MB+target, was added and allowed to sequester for 20 min at room temperature. For kinetics experiments, the solution containing the MB, either buffer or coacervate, was added to a fluorescence 384-microwell plate and then the target was added to each well using a multipipetter.

## Absorption spectra

Spectra were measured using a Cary 60 UV-Vis spectrophotometer for all samples. Measurements were performed within the spectral range of 200–900 nm, with 1 nm step intervals and a 0.1 sec integration time in a 10 mm path length quartz spectrophotometer cuvette was used for the measurements. Background scattering was corrected for using the OriginLab PeakAnalyzer function in the case of the coacervates samples.

## Fluorescence spectra

Spectra were measured at 20 °C using a TECAN Spark plate reader exciting from above on a 384-microwell plate. An excitation wavelength of 520 nm was used to excite the sample and the fluorescence emission was measured from 540–725 nm with 5 nm steps when collecting the entire spectra, or the Cy3 and Cy5 peaks were followed at 570 and 670 nm, respectively.

## Confocal fluorescence microscopy

Imagining was performed as reported previously[26]. Briefly, samples were imaged on functionalized glass slide–coverslip chambers[16]. 20 μl of sample was deposited in the chamber hole and covered with the coverslip. Imaging was performed using a Leica TCS SP8 STED ×3 with a ×60 objective lens (with oil immersion). Excitation was 520 nm with either the whole emission range collected (540–740 nm) for total intensity images or 10 nm windows, in the same range, were collected for spectral determinations.

## FRAP experiments

Measurements were performed as reported previously[26]. Briefly, Imaging was realized on a Marianas Spinning Disk confocal microscope (Intelligent Imaging Innovations) on a Zeiss Axio Observer inverted microscope using a ×100/1.46 NA PlanApochromat oil immersion objective. An area with radius = 0.5 μm was bleached for 5 ms with the 488-nm line from a solid state laser (LaserStack). Subsequent recovery of the bleached area was recorded with excitation from the 488-nm laser line and collected with a 440/521/607/700-nm quad emission dichroic and 525/30-nm emission filter. Images were acquired with a Prime sCMOS camera (Photometrics) controlled by SlideBook 6 (Intelligent Imaging Innovations).

## Data availability

The data that support the findings of this study are available from the corresponding author upon reasonable request. Supplementary information data 1 includes: MB absorbance spectra at varying temperatures, DNA melting curves (Fig. S1), absorbance spectra (Fig. S2), DNA sequestration quantification (Figs. S3 and S5), microscopy images (Fig. S4), FRAP (Fig. S6), random DNA sequence control (Fig. S7), and representative kinetic spectra (Fig. S8), additional FRET descriptions (Supplementary Note 1).

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

## Acknowledgements

The authors acknowledge the Office of Naval Research, the U.S. Naval Research Laboratory (NRL), and NRL Nanoscience Institute for programmatic funding, and S.A.D. acknowledged support from the OUSD RE in the form of a LUCI fellowship. R.V.U. acknowledges funding from Office of Naval Research for the Vannevar Bush Faculty Fellowship (grant N00014-21-1-2967). R.V.U., and D.S. thank the Air Force Office of Scientific Research (AFoSR) for funding of (grant FA9550-21-1-0091). D.M. was supported by the National Institute of Biomedical Imaging and Bioengineering of the National Institutes of Health under Award Number R00EB030013. The content is solely the responsibility of the authors and does not necessarily represent the official views of the National Institutes of Health.

## Author contributions

CMG, DS, JSM, PD, SAD: experimental results and sample preparation (investigation). CMG, DS, RU, SAD: data curation and formal analysis. CMG, DS, DM, SEG, ILM, RU, SAD: designed experimental protocols (methodology and conceptualization). SAD: original draft. All authors: writing—review & editing.

## Competing interests

The authors declare the following competing financial interest(s): a provisional patent, "Enhancing Fluorescence-Based Nucleotide Bio-sensors Through Liquid-Liquid Phase Separation Using Peptide-Based Coacervates", #63/594,358 USPTO, filed Oct. 30th, 2023, based on the technology has been filed by the U.S. Naval Research Laboratory and the authors (CMG, DS, ILM, RVU, SAD). All other authors declare no competing interests.
