## [Peer Review File · Communications Chemistry]

Reviewers' comments:

Reviewer #1 (Remarks to the Author):

The manuscript by Green et al. is an interesting study on how peptide coacervates improve the signaling response of molecular beacons. The authors show that the coacervates improve dye quantum yields, enhance kinetics, and substantially lower the detection limit.

Strengths: Overall, the work is an interesting bit of analytical and physical chemistry, and is a potential launching point for other studies how coacervates might improve the performance of fluorescent probes and sensors.

With exception of some minor comments below, the manuscript is well written, organized, easy to follow, and supported by clear figures and captions.

The work also has good novelty: Prior work by Abbas et al. (Nature Chemistry, 2021, 13, 1046) saw that a coacervate could take up and melt an RNA hairpin, but did not explore a hairpin system for target DNA detection.

Limitations: The authors discuss some limitations in the manuscript, which is quite useful. As addressed more directly in the comments below, the aspect that would benefit from some further consideration is the mechanism by which the fluorescence response is enhanced.

Recommendation: The manuscript reports on an interesting and novel phenomenon, is technically sound, and well presented. I recommend publication pending minor revisions.

Other Comments:

Ln. 34. Although I understood the term "bioindicators," the term "biomarker" is much more common.

Fig. 1B and Fig. 2. The spectra give the impression that the Cy3 fluorescence is modulating as expected for a molecular beacon, but that the Cy5 fluorescence is minimally variant regardless of whether the hairpin is open or closed, or in buffer or within the coacervate. Can the authors provide an explanation for this result?

In Fig. S3, the Cy5 fluorescence response appears non-monotonic and actually increases with the addition a sub-stoichiometric amount of target DNA. Is the trend unique to coacervate phase? If so, this data may suggest some sort of complex and/or cooperative behavior where the opening of one hairpin impacts the fluorescence of others. Can the authors provide any further explanation into what is occurring? Is this effect the origin of the remarkably lower limit of detection with the coacervate?

Ln. 71. Fig 1C is first figure call. Given explanatory role of Figure 1, it was somewhat awkward that 1A and 1B were not called in the text.

Ln. 90. What is pH of buffer? Please add.

Ln. 96. The authors should double-check the charge ratio for the coacervates. Unlike what is shown in Fig. 1C, ATP likely has a charge of 4- at the buffer pH and the R9 peptide will be 9+ or 8+ (depending on if the N-terminus is charged at the buffer pH). For 400 uM peptide and 750 uM ATP, the charge ratio will be closer to 6+:5- than 4+:3-.

The above noted, phases typically have a net neutral charge and counterions from buffer will presumably be recruited to balance any deficiency in ATP quantity. The authors should clarify that any charge effects in recruiting DNA to the coacervates are likely from charged hot spots. (Perhaps similar to Biplab et al., Chem. Sci. 2023, 14, 6608?)

Ln. 101. What was melt temperature of the hairpin? Please add to supplement the provided dsDNA melt temperature.

Table 2. QY measurements. While reading the main text, I was wondering about how scattering from the coacervates affected the quantum yield measurement using reference standards. In the Methods section (Ln. 332), a correction was noted in passing. How does a graphing program like Origin correct scattering? Explain the method or function.

Ln. 129 and Fig. S1. Why is the absorbance spectra for the open hairpin (i.e. dsDNA) not included? This data would clarify whether the strong coupling is due to the hairpin conformation AND the coacervate environment, or if strong coupling results from the coacervate environment regardless of a hairpin conformation. The extra peak around 600 nm is potentially suggestive of an H-aggregate for Cy5 or a large fraction of cis-isomer (instead of trans). Could aggregation or conformational effects have a role in the signal changes within the coacervate?

Does the coacervate have sufficient thermal stability to check for a change in melt transition for the hairpin within the coacervate environment? Experiments or discussion on this point would help address whether or not recruitment of target DNA to the coacervate phase was the main reason for the faster response.

Fig. 3. I recommend one significant figure on the uncertainty. Even if the curve fitting has precision to 0.04 ns, I doubt the experiment has precision to less than 0.1 ns.

Reviewer #2 (Remarks to the Author):

I read with interest this paper by Díaz et al., in which the authors present a proof of concept demonstrating that peptide coacervates based on polyarginine and ATP can serve as biosensors for DNA. The authors employ well-established and reasonably simple techniques, such as designing sequence-

specific molecular beacons, to develop a detection approach based on the FRET effect of cyanine dyes attached to the ends of DNA sequences. In my opinion, the work is well-presented, and the proposed concept is quite elegant given its simplicity. In this case, I recommend publication after a few clarifications are made as outlined below:

1) A central point of the work relies on sequestering DNA chains (both the target sequence and the molecular beacon) inside coacervates suspended in an aqueous phase. One concern is regarding the compositional homogeneity of the coacervates and the potential impacts of eventual heterogeneities on the reproducibility of the proposed biosensor. For instance, the authors state that "the continuous aqueous environment inside and surrounding the coacervate enables steady exchange between these phases." I fully agree with the statement, but differences in composition between coacervates suspended in the same solution could occur. In this case, potential heterogeneities could influence biosensing performance and make reproducibility challenging. Please elaborate a bit on this point.

2) The molecular beacon used for providing specificity contains 24 nucleotides, of which 16 correspond to the complementary sequence to the target sequence. The role of the other nucleotides in the MB is not clear to the reader, nor is it evident if the choice of these nucleotides was rationally designed. Please provide clarification.

3) The prepared coacervates likely exhibit liquid-crystalline characteristics, but this aspect is not discussed in the paper. The authors do highlight the high viscosity inside this phase as a preeminent factor for the increase in fluorescence quantum yield (which is correct). However, I think anisotropic domains within the coacervates could lead to the orientation of the fluorescent probe, possibly resulting in an elevated QY. I would like to see this aspect more thoroughly discussed by the authors and, if possible, the presentation of experimental evidence (e.g., polarized light microscopy) indicating whether or not there is anisotropy.

4) Line 130: The value of R_0 increases from 5.5 nm in the buffer to 6.8 nm inside the coacervates. The reason for this is not clear.

5) Figure 2: It is undeniable that the fluorescence intensity is substantially increased inside the coacervates. For example, comparing with the buffer, the intensities are 3 or 4 times higher. However, looking at the increase in QY values shown in Table 2, the increments in this parameter are generally much smaller. What then explains the observed increase in fluorescence?

6) I missed a hypothesis to explain the increase in the detection limit and sensitivity presented in the experiments shown in Figure 6. For example, in the case of the increase in QY, the authors attribute it to viscosity and the loss of the ability of the cyanine molecule to rotate, but for sensitivity (and LOD), only a description is provided without any mechanistic hypothesis. Please consider providing a mechanistic hypothesis to elucidate the underlying factors contributing to the increased sensitivity and lower limit of detection, thus offering a more comprehensive understanding of the experimental results.

Minor points:

Indicate the composition of the buffer in the Materials and Methods section. In the current version, the composition is only mentioned in the Results section.

Line 163: For the sake of reproducibility, please indicate the conditions used in the centrifugation in terms of relative centrifugation force (g).

Reviewer #3 (Remarks to the Author):

The manuscript by Diaz and coworkers reports that a molecular beacon (MB) partitioned into coacervate droplets (formed from ATP and a synthetic oligopeptide composed of 9 arginine units) functions as an efficient biosensor for a target oligonucleotide (DNA). The authors observe that lower ratios of target to MB are required to detect the target due to 1) concentration of the target in the coacervate droplets and 2) increased fluorescence intensity/quantum yield inside droplets. In addition, the authors also observe faster binding kinetics between MB and the target inside coacervates.

I believe this work will be of interest to a broad audience. The data are well-presented and the article is well-written. Use of multiple samples and/or multiple regions of an image to arrive at plotted profiles provides confidence that differences between diverse samples fall outside of error bars.

I fully support publication of this manuscript but prior to publication, it would be prudent to include one more control: a fluorescent image of coacervates containing everything (ATP, R9, target) except for the MB to confirm that any observed fluorescence is attributable to the MB in either hairpin or dsDNA conformation. Reference 31, from my brief perusal, does not appear to probe intrinsic fluorescence in the exact system used by the authors and it should be a quick and simple image to obtain for the paper.

A few other comments/questions I have are:

1. It appears that some of the fluorescent images are out of focus, is this something that could be improved?
2. The coacervate community is large and research in this area spans many decades. I would not refer to peptide coacervates as "a new class of biomaterial." In addition, it would be appropriate to reference some key works by the biophysics community, such as by the groups of Brangwynne and Papu, especially when discussing the internal structure of biomolecular condensates.
3. Presumably, in the setting where one is using such a platform as a biosensor there would be other biomacromolecules/molecules that could interfere with the analysis. Can the authors comment on this?

We thank all the reviewers for their positive feedback, insightful questions, and constructive criticism that we believe has led to an improvement of the manuscript quality. As can be seen from the new version of the manuscript we have endeavored to address all their concerns with modifications of the text and SI, including additional analysis, experiments, discussion, and references. Below we have aimed to answer the reviewer's questions and comments directly. Reviewer comments are in *Italic* with our response in bold or normal when highlighting portions from the main text.

Reviewer #1:

-Ln. 34. Although I understood the term "bioindicators," the term "biomarker" is much more common.

The text has been changed to biomarker.

-Fig. 1B and Fig. 2. The spectra give the impression that the Cy3 fluorescence is modulating as expected for a molecular beacon, but that the Cy5 fluorescence is minimally variant regardless of whether the hairpin is open or closed, or in buffer or within the coacervate. Can the authors provide an explanation for this result?

As noted in the text, this arises from the system changing from the hairpin version where the dyes are coupled to a less FRET efficient but much higher dye fluorescent QY system. We have aimed to improve the clarity of this fact in the main text, it now reads:

"We note that this strong coupling has been repeatedly shown to lower the fluorescence QY of cyanine dyes, including of heterodimer pairs as in this case.³¹ The proximity of specific nucleotides can also have an effect, as seen in the different dye QYs in the HP and dsDNA forms in Table 2..... Unexpectedly, we observed that the Cy5 signal increased upon going from the HP to dsDNA form (See Figure 2). This is due to the above-mentioned self-quenching of Cy5 by the vibronic coupling in the HP form.³⁰⁻³¹ Therefore, though the E_{FRET} is greatly reduced going from HP to dsDNA form, the fluorescence QY of both dyes improves to compensate for this issue."

-In Fig. S3, the Cy5 fluorescence response appears non-monotonic and actually increases with the addition a sub-stoichiometric amount of target DNA. Is the trend unique to coacervate phase? If so, this data may suggest some sort of complex and/or cooperative behavior where the opening of one hairpin impacts the fluorescence of others. Can the authors provide any further explanation into what is occurring? Is this effect the origin of the remarkably lower limit of detection with the coacervate?

To our understanding this question is similar to the above mentioned one and as such is already addressed.

-Ln. 71. Fig 1C is first figure call. Given explanatory role of Figure 1, it was somewhat awkward that 1A and 1B were not called in the text.

We have moved the figure up in the text and added specific call outs in the main text.

-Ln. 90. What is pH of buffer? Please add.

We thank the reviewer for catching this oversight. The 7.6 pH value has been updated in the text.

-Ln. 96. The authors should double-check the charge ratio for the coacervates. Unlike what is shown in Fig. 1C, ATP likely has a charge of 4- at the buffer pH and the R9 peptide will be 9+ or 8+ (depending on if the N-terminus is charged at the buffer pH). For 400 uM peptide and 750 uM ATP, the charge ratio will be closer to 6+:5- than 4+:3-.

Considering the 7.6 pH and that the final pKa of ATP is near this value, we used a 3.5 charge for the ATP, which results in the approximate 4:3 value we reported in the text.

-The above noted, phases typically have a net neutral charge and counterions from buffer will presumably be recruited to balance any deficiency in ATP quantity. The authors should clarify that any charge effects in recruiting DNA to the coacervates are likely from charged hot spots. (Perhaps similar to Biplab et al., Chem. Sci. 2023, 14, 6608?)

We thank the reviewer once more for catching this issue. The text has been modified and the pertinent reference has been included.

-Ln. 101. What was melt temperature of the hairpin? Please add to supplement the provided dsDNA melt temperature.

This was a good suggestion by the reviewer. The main text now includes the experimental determinations as well as the new data being included in the SI. The text now reads:

“The melting temperature of the HP form of the MB was determined to be 53 ± 1 °C in the buffer (coacervate values were not determined as the coacervate is not stable at these higher temperatures), while the dsDNA was determined to be 48 ± 2 °C (See Figure S1). As we worked consistently at 20 °C, we ensured that the initial structure was stable and that upon binding of the target it would not subsequently be released.”

-Table 2. QY measurements. While reading the main text, I was wondering about how scattering from the coacervates affected the quantum yield measurement using reference standards. In the Methods section (In. 332), a correction was noted in passing. How does a graphing program like Origin correct scattering? Explain the method or function.

The correction was only required for the absorbance spectra (a sentence clarifying that the software's Peak Analyzer tool was utilized has been added to the text). Due to the small optical path of the fluorescence measurements, it did not appear that the scattering modified the results.

-Ln. 129 and Fig. S1. Why is the absorbance spectra for the open hairpin (i.e. dsDNA) not included? This data would clarify whether the strong coupling is due to the hairpin conformation AND the coacervate environment, or if strong coupling results from the coacervate environment regardless of a hairpin conformation. The extra peak around 600 nm is potentially suggestive of an H-aggregate for Cy5 or a large fraction of cis-isomer (instead of trans). Could aggregation or conformational effects have a role in the signal changes within the coacervate?

The strong coupling is independent of the coacervate presence, though the particular configurations taken by the coupled dyes is likely unique to the environment. This is seen in the spectral shifts observed in the HP form in both the buffer and the coacervate in what is now Figure S2 in the SI. As confirmation, the figure has been updated and now includes the dsDNA form and we can see that the peaks align with the individual dye components.

-Does the coacervate have sufficient thermal stability to check for a change in melt transition for the hairpin within the coacervate environment? Experiments or discussion on this point would help address whether or not recruitment of target DNA to the coacervate phase was the main reason for the faster response.

This is an excellent question by the reviewer; unfortunately, the coacervate has a marked decrease in stability near the 35 °C mark, well below the DNA melting temperatures. A statement to that fact has been added in the discussion. The text now reads:

“Furthermore there are limits to temperature stability as well, for our system we noticed coacervate degradation (followed as a decrease in the scattering signal) at temperatures above 35 °C, with complete disappearance at 45 °C.”

-Fig. 3. I recommend one significant figure on the uncertainty. Even if the curve fitting has precision to 0.04 ns, I doubt the experiment has precision to less than 0.1 ns.

The figure has been modified upon reviewer suggestion.

Reviewer #2 (Remarks to the Author):

1) A central point of the work relies on sequestering DNA chains (both the target sequence and the molecular beacon) inside coacervates suspended in an aqueous phase. One concern is regarding the compositional homogeneity of the coacervates and the potential impacts of eventual heterogeneities on the reproducibility of the proposed biosensor. For instance, the authors state that "the continuous aqueous environment inside and surrounding the coacervate enables steady exchange between these phases." I fully agree with the statement, but differences in composition between coacervates suspended in the same

solution could occur. In this case, potential heterogeneities could influence biosensing performance and make reproducibility challenging. Please elaborate a bit on this point.

This is a valid point brought up by the reviewer. We have extended the discussion to point out the feasibility of their concern. Though specifically within our system we do not believe large heterogeneities can be found based on what we observed in the microscopy and FRAP experiments. The text now includes a paragraph that states:

“While we feel confident that the DNA is sequestered within the coacervates and our system appears homogenous based on our experimental evidence (microscopy and FRAP), anisotropic domains within the coacervate, liquid-crystal domains, or heterogenous probe distribution may be possible.^{11, 32, 39} There are examples of dye localization to the phase interface between the coacervate and the continuous buffer,²¹ and modified distributions could affect the biosensing response due to changes in local concentration or accessibility.”

2) The molecular beacon used for providing specificity contains 24 nucleotides, of which 16 correspond to the complementary sequence to the target sequence. The role of the other nucleotides in the MB is not clear to the reader, nor is it evident if the choice of these nucleotides was rationally designed. Please provide clarification.

We understand the reviewer’s desire for additional clarity, but we feel that the basis of MB design are explained within the cited references (particular ref#6). Specifically as concerns our system, the additional nucleotides are what cause the MB to fold into the HP structure in the absence of the target. We feel that between the description in the text and the schematic in Figure 1 and sequences in Table 1 that the system is adequately described.

3) The prepared coacervates likely exhibit liquid-crystalline characteristics, but this aspect is not discussed in the paper. The authors do highlight the high viscosity inside this phase as a preeminent factor for the increase in fluorescence quantum yield (which is correct). However, I think anisotropic domains within the coacervates could lead to the orientation of the fluorescent probe, possibly resulting in an elevated QY. I would like to see this aspect more thoroughly discussed by the authors and, if possible, the presentation of experimental evidence (e.g., polarized light microscopy) indicating whether or not there is anisotropy.

Though this is an interesting area of research, we feel it lies outside the scope of our current work. We have updated the text to refer to the reviewers comment and added a few pertinent references for any interested reader.

4) Line 130: The value of R0 increases from 5.5 nm in the buffer to 6.8 nm inside the coacervates. The reason for this is not clear.

We have updated the text to improve clarity. It now reads:

“For buffer the value is 5.5 ± 0.3 nm, while within coacervates it is 6.8 ± 0.3 nm, the increase being principally due to the increased QY of the Cy3.”

5) *Figure 2: It is undeniable that the fluorescence intensity is substantially increased inside the coacervates. For example, comparing with the buffer, the intensities are 3 or 4 times higher. However, looking at the increase in QY values shown in Table 2, the increments in this parameter are generally much smaller. What then explains the observed increase in fluorescence?*

We direct the reviewer to the values in Table 2 where the Cy3 values are increasing from 0.16/0.18 up to 0.52/0.62, exactly in the range of 3-4 times higher that the reviewer recognized in the figure.

6) *I missed a hypothesis to explain the increase in the detection limit and sensitivity presented in the experiments shown in Figure 6. For example, in the case of the increase in QY, the authors attribute it to viscosity and the loss of the ability of the cyanine molecule to rotate, but for sensitivity (and LOD), only a description is provided without any mechanistic hypothesis. Please consider providing a mechanistic hypothesis to elucidate the underlying factors contributing to the increased sensitivity and lower limit of detection, thus offering a more comprehensive understanding of the experimental results.*

We agree with the reviewer that additional analysis and discussion was warranted. Along with some minor changes in the text we have extended a section of the discussion which now reads:

“Yet, the functionality of DNA biosensors was not only preserved within coacervates, but they also worked at much lower concentrations and proportions of target strand to MB strand, due to the droplets capability to sequester oligonucleotides, raising their local concentration more than 100-fold. It is also possible that the R₉ destabilizes the MB allowing for a lower energy barrier for transition to the dsDNA form,³⁹⁻⁴⁰ improving kinetics and sensitivity. Other benefits included the improved S/N due to the increased fluorescence intensity of the dyes and enhanced kinetics. Though the full cause of the increased kinetics is not known at this time, we hypothesize that R₉ destabilization of the MB, along with the local increase in concentration maximizing collision frequency may play the two largest roles.”

Minor points:

-Indicate the composition of the buffer in the Materials and Methods section. In the current version, the composition is only mentioned in the Results section.

-Line 163: For the sake of reproducibility, please indicate the conditions used in the centrifugation in terms of relative centrifugation force (g).

The text has been updated with the buffer conditions in the MM section, as well as the inclusion of the buffer pH that reviewer 1 noted. The text has also been changed to RCF as per the reviewer’s suggestion.

Reviewer #3:

I fully support publication of this manuscript but prior to publication, it would be prudent to include one more control: a fluorescent image of coacervates containing everything (ATP, R9, target) except for the MB to confirm that any observed fluorescence is attributable to the MB in either hairpin or dsDNA conformation. Reference 31, from my brief perusal, does not appear to probe intrinsic fluorescence in the exact system used by the authors and it should be a quick and simple image to obtain for the paper.

The R₉-ATP coacervate system was chosen specifically as it has been well characterized in the past including by some of the current authors. We can direct the reviewer to the references by Fisher and Jain (#25 and #26) where no fluorescence was observed.

A few other comments/questions I have are:

1. It appears that some of the fluorescent images are out of focus, is this something that could be improved?

This is the result of the coacervates being found on multiple planes within the image. We could adjust the images through post-processing to increase sharpness, etc. We felt it best to show the images as we observed them without excessive post-processing.

2. The coacervate community is large and research in this area spans many decades. I would not refer to peptide coacervates as "a new class of biomaterial." In addition, it would be appropriate to reference some key works by the biophysics community, such as by the groups of Brangwynne and Papu, especially when discussing the internal structure of biomolecular condensates.

We agree with the reviewer and have modified the text to avoid giving this impression, as well as adding additional reference representative of this work.

3. Presumably, in the setting where one is using such a platform as a biosensor there would be other biomacromolecules/molecules that could interfere with the analysis. Can the authors comment on this?

The original text mentioned that high ionic strength or complex matrices could limit the applicability of the coacervates. It has now been updated with some additional statements clarifying that some experiments have been realized where the addition of enzymes were tested without modifying the results. We clarify to the reviewer that the enzymes we have tested are principally in the nuclease family, though we have not expanded on this fact in the manuscript, as this is still ongoing work.

The text now reads:

"We note that as part of a follow-up study we have tested the addition of enzymes to the mixture, from ~ 1-100 nM, and found no decrease in coacervate stability and efficiency (data not shown)."

REVIEWERS' COMMENTS:

Reviewer #1 (Remarks to the Author):

The authors have satisfactorily addressed my comments.

Reviewer #2 (Remarks to the Author):

The authors have adequately addressed my previous comments by adding further clarifications and discussions. I am satisfied with the responses. In my opinion, the current version is suitable for publication.

Reviewer #3 (Remarks to the Author):

The points raised in the first round of revisions were appropriately addressed and I support publication of the manuscript as is.